# A Time-Lagged Examination of the Greenhaus and Allen Work-Family Balance Model

**DOI:** 10.3390/bs10090140

**Published:** 2020-09-18

**Authors:** Alfonso Landolfi, Massimiliano Barattucci, Alessandro Lo Presti

**Affiliations:** 1Dipartimento di Psicologia, Università degli Studi della Campania “Luigi Vanvitelli”, Viale Ellittico, 31-81100 Caserta, Italy; alessandro.lopresti@unicampania.it; 2Facoltà di Psicologia dell’Università e-Campus, via Isimbardi, 10-22060 Novedrate (CO), Italy; massimiliano.barattucci@uniecampus.it

**Keywords:** work-family balance, work-family enrichment, work-family conflict, satisfaction, demands, resources, core self-evaluations

## Abstract

The work-family interface is a compelling topic that calls into question labor market dynamics and work processes, together with important social and family composition changes. The present study aimed at examining the antecedents of Work-Family Balance (WFB) in Italy consistent with Greenhaus and Allen’s (2011) conceptual model in which the characteristics of work and family roles have an indirect impact on work-family balance through Work-Family Conflict (WFC) and Work-Family Enrichment (WFE), and where job and family satisfaction are considered as predictors of WFB. A total of 568 workers participated in a time-lagged correlational study, filling a questionnaire. The theoretical model was tested by assessing the mediating role of job and family satisfaction as well as related antecedents, conflict, and enrichment between the family and work contexts, through Structural Equation Modeling (SEM). The results partially confirmed the theoretical model: work-to-family enrichment and work-to-family conflict predicted family satisfaction, which also mediated their association with WFB. The results in the family-to-work direction did not support the initial research hypotheses. The hypotheses about associations between demands and resources, conflict and enrichment in both directions, and of the moderating role of core self-evaluations were partially confirmed. The results highlighted that organizations need to carry out periodic assessments of WFC and WFE, in order to provide benefits and resources, to reduce conflict, and increase enrichment, through proper interventions (training activities, professional development, mentoring, and forms of flexibility).

## 1. Introduction

The work-family interface appears to be a compelling and challenging topic that can be interpreted as the result of important changes (also of a cultural and social nature) regarding the relationship between family and work domains [1,2,3]. Another critical aspect is that, nowadays, workplaces still seem largely designed on the basis of a culture in which work plays a pivotal role in employees’ lives and the belief that workers should sacrifice family roles in order to be successful in the workplace [4].

Questions can be raised on how organizations, workers, and their partners can establish a balance between family and work domains. The balance between family and work demands became a challenge that falls within a general dimension having to tackle labor market dynamics and work processes, together with important social and family composition changes [3,5]. Within this context, employers and employees alike are becoming increasingly more interested in measures that promote the establishment of a balance between work and family [6]. Therefore, it deserves to be paid attention from different standpoints also due to the important consequences it has, not only on individual and family health and well-being [6,7] but especially on organizations (e.g., absences from work, leave for family assistance, sick days, turnover, etc.) [6,8] and on welfare policies.

Two different views can be found regarding the concept of Work-Family Balance (hereafter, WFB). On the one hand, WFB is conceptualized as a distinct construct from Work-Family Conflict (hereafter, WFC), Work Family-Enrichment (hereafter, WFE), or Work Family-Facilitation [9,10,11]. On the other hand, other scholars argue that the concepts of WFC and WFE are essential to capture the multi-dimensionality of WFB [12,13]. Unlike WFB, WFC and WFE can be of a bi-directional nature (from work-to-family and from family-to-work). Some contributions evaluated WFB as a result of a low level of WFC or at least a combination of low work-family conflict and high work family-enrichment [10], while for others WFB is a multi-dimensional construct consisting of both WFC and WFE [13]. Greenhaus and Allen developed a conceptual model in which the characteristics of the work and family roles can have an indirect impact on WFB through WFC and WFE [9] and job and family satisfaction is conceived as a predictor of WFB, as far as is known, few other studies have examined this relationship according to the direction pursued by Greenhaus and Allen [14].

WFB is positively related to job and family satisfaction, including well-being, and negatively related to anxiety and depression [6,7,15,16,17,18], so a balanced investment of time and involvement in work and family would certainly reduce conflicts and work-family stress, thus improving the general quality of life [6]. Some indications have underlined that scholars examining WFB should pay attention to the role of dispositional variables since some personality traits seem to influence the relationship between work and family domain [13,19].

Since the literature on WFB still looks controversial and has received limited empirical attention [8], a correlational research was designed with the aims of examining and testing, within the Italian context, the model by Greenhaus and Allen [9]; highlighting the differential and indirect effects of conflict and enrichment on WFB through job and family satisfaction; and analyzing the moderating role of a dispositional factor, namely Core Self-Evaluations (hereafter, CSE). From a practical standpoint, improving their knowledge of WFB dynamics would allow organizations to be able to support and help workers to find a greater balance through specific organizational services and interventions, which would be reflected not only on well-being, individual and family satisfaction but also on performance, commitment, turnover intentions, and job satisfaction [8,9].

## 2. Theoretical Foundation

Several WFB-related studies have been carried out and most of them analyzed its relationship with both work-family conflict and enrichment, as well as with job- and family-satisfaction [8,18,20,21]. Greenhaus and Allen [9] developed a conceptual model in which the characteristics of work and family roles can have an indirect impact on WFB perceived through WFC and WFE, and considered, unlike other studies, job and family satisfaction as predictors of WFB (see Figure 1).

WFB was defined as “an overall appraisal of the extent to which individuals’ effectiveness and satisfaction in work and family roles are consistent with their life values at a given point in time”. Job and family resources can reduce WFC and increase WFE, and these two bi-directional dimensions (from work-to-family and from family-to-work) can, in turn, promote effectiveness and satisfaction in both work and family roles, which are the closest antecedents of the WFB: feelings of balance are produced by an interaction of effectiveness and satisfaction with life values.

These can significantly impact important individual and work dimensions: the quality of life [6], organizational commitment, and family performance [8], as well as well-being [6,15,16,17]. Likewise, as for WFB antecedents, scholars found evidence for the predictive value of, among others, family-supportive supervision [10], career attitudes [22], co-workers, and partner support [23]. Demands and resources alongside job and family spheres can provide individuals with valuable psychosocial resources that can facilitate their subsequent work experience. Overall, research on the work-family interface is characterized as if it was organized in two directions, work-to-family and family-to-work.

### 2.1. Work-To-Family Experiences

The experiences of conflict and enrichment play a pivotal role in balance studies. The work-to-family direction refers to the WFC and WFE constructs. In general, this direction explains how work experiences can have either negative or positive effects within the family domain, just as the pressures, demands, or resources generated within the job context interfere with the activities in the family context. Both constructs have been widely studied as a form of negative (conflict) and positive (enrichment) work-family interaction.

The first researchers to systematize the WFC dimension were Greenhaus and Beutell [24]; they defined WFC as “a form of inter-role conflict in which the role pressures from the work and family domains are mutually incompatible in some respect” [24] (p. 77). Instead, Greenhaus and Powell defined WFE “as the extent to which experiences in one role improve the quality of life in the other” [25] (p. 73). WFE is conceptually and empirically distinct from WFC [26], in fact, WFE is not the simple absence of the other: an individual may well experience high levels of both conflicts and enrichment/positive spillover simultaneously.

Job resources and demands seem to play an important role as antecedents of WFC and WFE: they represent working conditions that concern different job aspects such as workload (i.e., job demands), job control, co-workers, and supervisor support (i.e., job resources). While the former requires mental and physical effort associated with a process of compromising health (of a psychological or physiological nature), the latter helps with managing stressful situations and is functional to the achievement of work objectives [27].

As for WFC antecedents, demands related to different roles [24,28,29]—dispositional characteristics (Type A personality, locus of control) [11], and work time [29]—may be mentioned. In regards to job resources, Bellavia [30] found that support provided by supervisors and co-workers could decrease WFC, while other researchers [24,31,32,33] examined the link between demands and resources, and other job characteristics (e.g., hours worked, inflexible work schedule, reduced or part-time work hours for parents, co-worker support, workload, etc.) and WFC. Michel, Kotrba, Mitchelson, Clark, and Baltes [34] highlighted that social support received in the workplace (and within the family context) was negatively associated with conflict.

Bhargava and Baral [19] identified supervisor support and other job characteristics (e.g., job autonomy and variety), while Lo Presti et al. found work-family organizational support to be an antecedent of WFE [21]. Hence, social support is an important antecedent of both directions (FWE and WFE), and co-worker support and workload seem to be fundamental variables that can either favor or hinder FWE development [35,36]. In short, while resources can either facilitate or enrich their subsequent job experiences [37], demands, on the other hand, can prevent and contrast enrichment, in turn generating conflict between the job and family domains. Summing up, consistent with the Greenhaus and Allen model on WFB [9], it can be expected that:

**Hypothesis** **1** **(H1).**
*job resources will be negatively related to WFC (a) and positively related to WFE (b);*


**Hypothesis** **2** **(H2).**
*job demands will be positively related to WFC (a) and negatively related to WFE (b).*


### 2.2. Family-to-Work Experiences

The potential negative and positive effects of participation in one life domain (either work or family) on performance in another life domain (either family or work) are generally recognized in literature [9,25]. Just the same as WFC and WFE, there are two dimensions in the family-to-work direction as well: FWC and FWE. In this direction, the role pressures, demands, or resources generated within the family context interfere with the work domain (i.e., FWC). Whereas, FWE makes it possible to highlight how the energy and/or involvement generated in the family domain helps the individual to effectively manage the activity in the job domain.

An important role, as antecedents, is played by family resources and demands. These are working conditions that concern different family aspects such as family workload (i.e., a family demand) and family social support (i.e., a family resource). As for family resources, it has been shown that they have a predictive role with respect to enrichment [38], as well as family support that seems to have fundamental variables that can either favor or hinder the FWE development [35]. Likewise, Bhargava and Baral [19] identified family support as an antecedent of WFE, while in a meta-analytical review [34], family role, overload family stressors, and family demands, together with supportive antecedents such as spousal support and familiar climate, were found to have significant effects on the conflict. Summing up, it was expected that:

**Hypothesis** **3** **(H3).**
*family resources will be negatively related to FWC (a) and positively related to FWE (b);*


**Hypothesis** **4** **(H4).**
*family demands will be positively related to FWC (a) and negatively related to FWE (b).*


### 2.3. Work-Family Balance, Enrichment and Conflict alongside Satisfaction

Conflict and enrichment have, respectively, several negative and positive consequences on the psychological and organizational outcomes [39,40]), while both WFC and WFE predict WFB [41]. As for WFC, a negative relationship with WFB was shown in recent studies [8,10,20]. For WFE, Lo Presti et al. [21] found that the improvement of the quality of life in one role (work) in consequence of experiences in the other role (family) lead to a beneficial influence on WFB. Other studies have shown that there is a positive relationship between WFE and WFB [8,10,20]. Several studies showed that both directions (WFE and FWE) have a positive relationship with job and family satisfaction [13,42,43,44,45]; moreover, family satisfaction, job satisfaction, affective commitment, and organizational citizenship behavior (OCB) were proven to be predictable by WFE [19].

As for conflict outcomes, various studies showed that there is a negative relationship, in both directions, regarding job and family satisfaction [39,42,46,47]: conflict has negative effects on behavior and well-being and is considered to be a potential source of stress. According to Kossek and Ozeki [48], conflict plays a significant role in determining dissatisfaction with both work and life.

Other factors, besides conflict and enrichment, may also influence WFB. Among these factors, in line with Greenhaus and Allen model [9], a particular role is played by satisfaction. Job and family satisfaction are generated by an assessment, carried out by individuals, regarding their work and family lives or situations, characterized by more or less pleasing emotional states [49].

Several studies have shown positive associations between WFB and job and family satisfaction [18,20,21,50]. Greenhaus and Allen [9] advanced a different viewpoint, suggesting that the sensation of balance between family and work roles is generated only when individuals feel very satisfied and effective in their respective roles, while poorer satisfaction would cause an imbalance. Therefore, it was expected that:

**Hypothesis** **5** **(H5).**
*family satisfaction will be negatively predicted by WFC (a) and will be positively predicted by WFE (b);*


**Hypothesis** **6** **(H6).**
*job satisfaction will be negatively predicted by FWC (a) and will be positively predicted by FWE (b);*


**Hypothesis** **7** **(H7).**
*WFB will be positively associated with family (a) and job satisfaction (b).*


### 2.4. The Role of Dispositions

Personality factors play a pivotal role in various contexts of life, determining the emotional and behavioral adaptation of individuals to their environments, as well as their job attitudes and work-related behavior [51,52]. Several work-family interface studies examined the role of individual differences showing that certain personality traits are associated with enrichment, such as greater extroversion [1,53]. Likewise, other dispositional traits impact on conflict, such as neuroticism (or low emotional stability) [11,54,55]. Similarly, in a meta-analytical review of the antecedents of WFC, Michel, Kotrba, Mitchelson, Clark, and Baltes [34], showed the significant effect of personality variables within the family-work context (including the locus of control and negative affect/neuroticism). This evidence suggests that dispositional variables seem to play a crucial role within the work-family context.

An integrated construct in the work-family area is represented by CSE, which was introduced by Judge et al. [51] in an attempt to explain employee attitudes and behavior, and was defined as “the fundamental assessments that people make about their worthiness, competence, and capabilities” [56] (p. 257). This dimension includes four personality dimensions: self-esteem, neuroticism, locus of control, and general self-efficacy, representing [57] the assessments that people carry out not only of themselves but also of other people and the world in general.

Several studies show that CSE is a reliable predictor of important workplace outcomes [57,58]. Along this line, Boyar and Mosley [59] elaborated a model that integrated CSE into the work-family context and that incorporated WFC and work-family facilitation (or enrichment) variables together with job satisfaction and family satisfaction. CSE emerged as a significant predictive factor in relation to enrichment [19,59]. In fact, according to Jain and Nair [35], CSE was found positively related to family-to-work enrichment (FWE).

Based on the abovementioned arguments and on the few pieces of evidence reported in the literature, and in line with the theoretical framework used, it is expected, in an exploratory way, that CSE can moderate the relationship between WFC and WFE with family and job resources and demands. In particular, consistent with the Greenhaus and Allen WFB model [9], CSE is likely to act as a moderator between both family and job resources as well as demands and experiences relating to the conflict and enrichment between family and job contexts.

## 3. Method

### 3.1. Participants and Procedure

The participants were 601 Italian employees, voluntarily recruited through a convenience sampling strategy, contacted within organizations by trained researchers. In this research, both the Helsinki Declaration [60] and the Italian laws of data protection (legislative decree n.196/2003) were adhered to and all study participants provided their informed consent.

At Time 1, individuals filled out a paper-and-pencil self-report questionnaire of 60 items. The first page of the questionnaire contained the objectives of the study, the instructions for participation, and a declaration on data processing in compliance with current Italian laws. After about three months, the participants were asked to fill in a second questionnaire of 21 items (Time 2). This time-lagged design surveyed the job and family resources and demands, WFC, WFE, FWC, and FWE at Time 1, as well as job satisfaction, family satisfaction, and WFB at Time 2. Further, different formats and scale endpoints were used in order to reduce method biases caused by commonalities in scale endpoints and anchoring effects. In total, 576 questionnaires were returned.

A subsequent and more in-depth evaluation allowed to remove further eight cases because the questionnaires were only partially completed. The final sample consisted of 568 workers.

About gender, 263 (46.3%) were men and 305 (53.7%) were women, with the age range being between 17 and 67 years. Their average number of children was 1.45 (SD = 1.06). Regarding the educational levels, 137 (24.1%) held an elementary/junior high school certificate, 238 (41.9%) a high school diploma, and 193 (34%) a university degree or higher.

As for marital status, 96.1% of the participants were married or cohabiting, while regarding the employment status, 469 participants (82.6%) had a permanent employment contract, while 54 (9.6%) had a fixed-term/temporary contract, 44 (7.7%) held other statuses There was one missing value (0.1%). The average organizational tenure was 14.66 years (SD = 11.27), while their average total tenure in their entire career was 21.74 years (SD = 12.17). There were 177 (31.2%) blue-collar workers, 314 (55.3%) were white-collars, while 54 (9.5%) were managers, and 21 (3.7) were professionals or self-employed (two missing values). Finally, 24 workers (4.2%) were employed in the primary sector, 177 (31.82%) in the secondary one, while 367 (66.4%) worked in the tertiary one. Table 1 summarizes the demographic variables’ descriptive statistics.

### 3.2. Measures

All measures were taken from validated questionnaires, and all of them were already available in Italian.

#### 3.2.1. Job Demands

The *Workload* measure [61] (Italian version [62]) included three items (e.g., “Do you have too much work to do?”). Responses were based on a five-point frequency scale (from 1 = never to 5 = always), and scores equal to the mean of the three items. Cronbach’s alpha was 0.84.

#### 3.2.2. Job Resources

The *Coworkers support* measure [61] (Italian version [62]) included three items (e.g., “Can you count on your colleagues when you face difficulties at work?”). Responses were based on a five-point frequency scale (from 1 = never to 5 = always), scores equal to the mean of the three items. Cronbach’s alpha was 0.81. The *Supervisor support* measure [26] (Italian version [62]) included three items (e.g., “My supervisor and I get along well”). Responses were based on a five-point frequency scale (from 1 = never to 5 = always), scores equal to the mean of the three items. Cronbach’s alpha was 0.91.

#### 3.2.3. Family Demands

*Family workload* [38] (short version) was assessed through six items that asked to evaluate the frequency at which the individual was in charge of accomplishing a series of family chores within their own household (e.g., “do grocery shopping”, “prepare a hot meal”, etc.). Responses were based on a five-point frequency scale (from 0 = never to 4 = always) while scores ranged between 0 and 32. Cronbach’s alpha was 0.94.

#### 3.2.4. Family Resources

*Emotional family social support* was assessed via six items (e.g., “Members of my family are interested in my job”) from the shortened version [38] of the Family Support Inventory by King, Mattimore, King, and Adams [63], and refers to the amount of emotional support perceived by the respondent. Participants used a five-point scale ranging from 1 = completely false to 5 = completely true. Cronbach’s alpha was 0.80.

*Instrumental family social support* was assessed via six items (e.g., “My family leaves too much of the daily details of running the house to me”) from the shortened version [38] of the Family Support Inventory by King et al. [63], and refers to the amount of instrumental support perceived by the respondent. Participants used a five-point scale ranging from 1 = completely false to 5 = completely true. Cronbach’s alpha was 0.81.

#### 3.2.5. Disposition

The *CSE* scale [64] (Italian version [65]) was composed of 12 items (e.g., “When I try, I generally succeed”), with response options presented in a five-point Likert scale format ranging from strongly disagree (1) to strongly agree (5). Cronbach’s alpha was 0.80.

#### 3.2.6. Mediators

The *Work-to-family conflict* measure [66] (Italian version [67]) included five items (e.g., “The demands of my work interfere with my home and family life”) with a Likert scale from 1 = completely disagree to 7 = completely agree. Cronbach’s alpha was 0.90.

The *Family-to-work family conflict* measure [66] (Italian version [67]) included five items (e.g., “Things I want to do at work do not get done because of the demands of my family or spouse/partner”) with a Likert scale from 1 = completely disagree to 7 = completely agree. Cronbach’s alpha was 0.86.

The *Work-to-family enrichment* measure [26] (Italian short version [68]) comprised three items (e.g., “At work, I develop new skills and this helps me to be a better family member”) with a five-point Likert scale from 1 = completely disagree to 5 = completely agree. Cronbach’s alpha was 0.87.

The *Family-to-work enrichment* measure [26] (Italian short version [68]) comprised three items (e.g., “In my family life I develop new skills and this helps me to work better”) with a five-point Likert scale from 1 = completely disagree to 5 = completely agree. Cronbach’s alpha was 0.76.

The *Family satisfaction* measure [69] (Italian version [21]) refers to the extent to which the respondent is satisfied with his/her own family life and was assessed through five items (e.g., “In most ways my family life is close to my ideal”) with a seven-point Likert scale ranging from 1 = completely disagree to 7 = completely agree. Cronbach’s alpha was 0.96.

The *Job satisfaction* measure [70] (Italian version [71]) comprised five items with a five-point scale from 1 = very unsatisfied to 5 = very satisfied (e.g., “Indicate your satisfaction about…physical working conditions”). Cronbach’s alpha was 0.92.

#### 3.2.7. Outcome

The *Work-family balance* measure [8] (Italian version [72]) comprised six items, (e.g., “I am able to negotiate and accomplish what is expected of me at work and in my family”) with a five-point Likert scale from 1 = completely disagree to 5 = completely agree. Cronbach’s alpha was 0.94.

### 3.3. Data Analysis

Missing values (.004% of all expected cells for Time 1 scales, 0.003% for the Time 2 ones) were replaced through the Expectation–Maximization method (SPSS 21) [73]. Descriptive statistics and correlations were calculated through IBM SPSS 21 in order to investigate associations between variables.

We used, to evaluate the measurement and structural models concerning the study variables under interest and their associations, structural equation modeling analyses (Lisrel 9.3) using the Robust Maximum Likelihood estimation method (along with the indicators’ covariance matrix). Since the sample size was small to comply with the rule of at least 10 cases for each parameter to be estimated [74], to get more precise parameter estimates, increased reliability, a better model fit, and less biased estimates, we relied on item parceling for estimating latent constructs, and reduced levels of skewness and kurtosis [75,76].

We computed parcels for Time 1 and Time 2 scales after an exploratory factor analysis (estimation method: principal axis factoring) and comprised items aggregating those with the highest and lowest loadings.

Job resources (two parcels: supervisor support and coworkers support), family resources (two parcels: instrumental family social support and emotional family social support), family demands (three parcels each including two items of the family-workload scale), work-to-family conflict (parcel #1: items 2 and 4; parcel #2: items 3 and 5; parcel #3: item 1), family-work conflict (parcel #1: items 1 and 4; parcel #2: items 3 and 5; parcel #3: item 2), job satisfaction (parcel #1: items 1 and 4; parcel #2: items 2 and 5; parcel #3: item 3), family satisfaction (parcel #1: items 2 and 4; parcel #2: items 3 and 5; parcel #3: item 1), work-family balance (parcel #1: items 2 and 6; parcel #2: items 4 and 5; parcel #3: items 1 and 3).

Fit indices that minimized the likelihood of Type I and Type II errors [77] were selected: the chi-square test (χ^2^), the Comparative Fit Index (CFI), the Non-Normed Fit Index (NNFI), the Standardized Root Mean Residual (SRMR), and the Root Mean Square Error of Approximation (RMSEA; with 95% confidence interval lower and upper limits, hereafter 95% CI [LL, UL]). Criteria for the goodness of these fit indices can range from less (CFI, NNFI ≥ 0.90; SRMR, RMSEA ≤ 0.10) to more conservative criteria (CFI, NNFI ≥ 0.95; SRMR, RMSEA ≤ 0.08; [76]).

In order to test for the possible moderating role of CSE, multi-group structural equation modeling was used [78]. A first model with all parameters invariant between groups was compared with an alternative model wherein each parameter (i.e., gamma) at a time was released (i.e., free across groups). An inferential test about the χ^2^ difference between the two models with one degree of freedom was then carried out in order to verify significant differences between both groups (moderator’s low scores versus moderator’s high scores).

## 4. Results

A measurement model was developed in order to examine the construct validity of study measures using Confirmatory Factor Analysis (CFA); a common method is to compare two models (nested models): a one-factor model and a final one containing as many factors as the included measures (in our case 11 latent variables). The two models were compared on the basis of chi-square/degrees of freedom scores, and on different goodness of fit indices (Table 2).

It is possible to notice a remarkable improvement in all goodness of fit indexes of Model 2 (complete) compared to Model 1 (one factor). In particular, Model 2 showed satisfactory goodness of fit indexes (χ^2^ = 3656, df = 1188, CFI = 0.93, SRMR = 0.06, RMSEA = 0.059, NNFI = 0.92) providing support for construct validity of all study variables.

### 4.1. Descriptive Findings

Table 3 depicts all study variables’ descriptive statistics and zero-order correlations.

Workload negatively correlated with work-family enrichment (*r* = −0.12, *p* = 0.004), while Coworker support (*r* = 0.13, *p* = 0.001), supervisor support (*r* = 0.30, *p* < 0.001), general family support (*r* = 0.23, *p* < 0.001) positively correlated with work-family enrichment. Work-family conflict (*r* = −0.09, *p* = 0.02), family-work conflict (*r* = −0.12, *p* = 0.003), and family workload (*r* = −0.16, *p* < 0.001) negatively correlated with work-family balance, while work-family enrichment (*r* = 0.09, *p* = 0.02), supervisor support (*r* = 0.09, *p* = 0.03), core self-evaluations (*r* = 0.26, *p* < 0.001), job satisfaction (*r* = 0.57, *p* < 0.001), family satisfaction (*r* = 0.59, *p* < 0.001), and general family support (*r* = 0.10, *p* = 0.01) positively correlated with work-family balance.

### 4.2. Direct and Indirect Associations

We tested the hypothesized direct and indirect relationships through a structural model. The estimated model showed adequate goodness of fit indices: χ^2^ = 1114.29 df = 418, CFI = 0.97, GFI = 0.89, SRMR = 0.1, RMSEA = 0.048 CI (.044; 0.052), NNFI = 0.96, *p*-Value for Test of Close Fit (RMSEA < 0.05) = 0.78. Figure 2 depicts the final structural model.

WFC was positively predicted by job demands (*β* = 0.28, *p* < 0.001) and negatively predicted by job resources (*β* = −0.12, *p* = 0.005), while WFE was positively predicted by job resources (*β* = 0.26, *p* < 0.001) and negatively predicted by job demands (*β* = −0.11, *p* = 0.005). Family resources negatively predict FWC (*β* = −0.31, *p* < 0.001) and positively predict family-work enrichment (*β* = 0.43, *p* < 0.001).

As regards family demands, their positive relationship with FWC was not significant (β = 0.04), while, counterintuitively, their relationship with FWE was significant and positive (β = 0.11, *p* = 0.005). The results of this latter association are counterintuitive with respect to previous studies [9]. Moreover, WFC negatively predicted family satisfaction (*β* = −0.09, *p* = 0.05), likewise family satisfaction was positively predicted by WFE (*β* = 0.14, *p* = 0.005), while the relationships of FWE and FWC with job satisfaction were both not significant (respectively β = 0.08 and β = −0.07). In turn, job satisfaction positively predicted WFB (*β* = 0.47, *p* < 0.001), as well as family satisfaction (*β* = 0.41, *p* < 0.001).

Again, significant indirect effects of WFE towards WFB through family satisfaction were found (β = 0.07, *p* = 0.01), while the same indirect effects but negative, were found for WFC towards WFB (β = −0.08, *p* = 0.05), always through family satisfaction.

As for the explained outcome variables’ variance, the predictors in this research model explained a significant amount of variance in WFB (48%, *p* < 0.001), in WFE (8%, *p* < 0.001), and in FWE (20%, *p* < 0.001). Regarding FWC and WFC, the amount of explained variance was the same for both (10%, *p* < 0.001). While the amount of explained variance for satisfaction was significant for family satisfaction (2%, *p* = 0.05), but not (1%) for job satisfaction.

### 4.3. Multivariate Associations as Moderated by Dispositional Trait (CSE)

First, a model with all parameters set as invariant between groups (CSE’s low scores versus CSE’s high scores) was computed and served as the baseline model for subsequent comparisons. The estimated model showed adequate goodness of fit indices: χ^2^ = 1991.42, df = 907, CFI = 0.95, NNFI = 0.95, RMSEA = 0.055 95% CI [0.050, 0.059], *p*-Value for Test of Close Fit (RMSEA < 0.05) = 0.066.

Then, several alternative structural models, each time with a single parameter (gamma) left to be free between the two groups, were estimated. From the comparison of these data, only one model emerged indicating an association between variables that differed significantly between groups. This model (χ^2^ = 1981.96, df = 906, CFI = 0.95, NNFI = 0.95, RMSEA = 0.054 95% CI [0.050, 0.058]), whose χ^2^/df difference = 9.97 with the baseline model, was significant at *p* = 0.002, and showed that the association between family resources and family-work conflict differed between the two groups in particular, the association was negative and significant both among CSE’s low (β = −0.41, *p* < 0.000), and high scores (β = −0.14, *p* = 0.05).

Finally, single structural models CSE’s low scores group (χ^2^ = 947.99, df = 418, CFI = 0.96, NNFI = 0.95, SRMR = 0.09, RMSEA = 0.055 95% CI [0.049, 0.061], Value for Test of Close Fit [RMSEA < 0.05] = 0.060), and CSE’s high scores group (χ^2^ = 712.38, df = 418, CFI = 0.97, NNFI = 0.97, SRMR = 0.09, RMSEA = 0.041 95% CI [0.033, 0.058], *p*-Value for Test of Close Fit [RMSEA < 0.05] = 0.99) separately, were tested. Summing up, only one CSE moderation effect was found to be significant. We found that CSE moderated the negative relationship between family resources and family-to-work conflict, this showed that this negative relationship was weaker for individuals reporting higher scores of CSE and was stronger for individuals reporting lower scores of CSE (low: *β* = −0.41, *p* < 0.001; high: *β* = −0.14, *p* = 0.05). In all, CSE attenuates the negative association between family resources and FWC.

## 5. Discussion

The main purpose of this study was to examine WFB’s antecedents in Italy, in line with the model developed by Greenhaus and Allen [9] through a time-lagged design in order to avoid the common method bias and provides a more rigorous test for non-spurious associations than cross-sectional studies.

According to these authors, WFB is the outcome of family and job satisfaction, which, in turn, are related to conflict and enrichment processes (in both directions: work-to-family and family-to-work). An attempt was made to identify the proximal antecedents of WFB, such as job and family satisfaction, as well as the several factors that influenced WFE and WFC, in turn strongly associated constructs directly and indirectly linked to WFB [8,9,10,20]. Moreover, this study examined if the hypothesized associations between variables may be different in relation to CSE since in the work-family literature, personality factors are salient [35].

Firstly, this study supported previous evidence [8,9] concerning the difference between WFB, WFC, and WFE. In line with our hypotheses and with what has already been reported in the literature [9,14], both WFC (negatively) and WFE (positively) have indirect effects on the WFB.

Overall, the work-to-family direction has shown a greater impact on the feeling of balance than the family-to-work direction. Family resources showed an association with both enrichment processes and conflict, just as job demands also showed a significant association with WFC and WFE, respectively positively and negatively. Furthermore, both job and family satisfaction were significantly associated with WFB.

As for our hypotheses, the results have shown that WFE is associated with higher family satisfaction, while WFC is negatively associated with family satisfaction. As for the family-to-work direction, contrary to our hypotheses, both FWC and FWE have no significant effect on job satisfaction.

An important and critical aspect concerning this study reveals that both job satisfaction and family satisfaction are associated with a higher WFB, consistent with Greenhaus and Allen [9]. This aspect means that satisfaction could not only improve the balance between family and work contexts but could also affect some of the positive and negative results both for the family (e.g., well-being in the family or family stress) and for the organization (e.g., organizational stress or organizational involvement or commitment).

As regards the potential predictive role of job demands and resources, alongside family demands and resources, given that they are present in the workplace and they cannot be eliminated, the potential for CSE to improve or minimize/buffer the impact of these resources or demands was examined.

Overall, results suggest, also in the wake of other studies [8,36], that job and family resources and job and family demands play an essential role in managing the dynamics relating to family and work contexts; in fact, they can contribute to counterbalancing, also indirectly, the perception of incompatibility and conflict, improving both enrichment and feelings of balance generated in the aforementioned contexts.

According to the results, the main aspect of this study concerns the importance that family satisfaction and job satisfaction have as factors establishing a sensation of balance. In particular, family satisfaction is able to mediate the effects deriving from conflict and enrichment episodes in the work-to-family direction.

Moreover, given the importance of predictive factors, future studies should include other determinants in order to examine their predictive power to find out which additional job resources can promote the WFB or which demands need to be kept under control to avoid a poor balance between job and family domains. Table 4 summarizes direct effects in relation to the study hypotheses, while Table 5 summarizes the significance of moderated effects by CSE.

### 5.1. Study Limitations

Although our study has several theoretical and practical implications, there are some limitations to consider when interpreting our results.

Firstly, although our research design is time-lagged, given the nature of the hypotheses of this study, causal inferences cannot be made regarding the relationships between the variables. Given the complexity of the model, in order to overcome this limitation, future research should preferably focus more on cross-lagged or diary studies.

Another aspect to mention concerns the nature of sampling: the convenience sampling procedure does not enable the sample to be considered as a representative of all couples, which prevents strong inferences of generalizability to be made to the wider population.

Another limitation of the present study in relation to the Greenhaus and Allen model concerns the limited importance attributed to individual differences, in fact, it is plausible to expect that different additional dispositional variables may moderate the impact of the WFB both with respect to its antecedents and respect to its outcomes.

As regards paths for future studies, further scholarly effort is required also with regards to other aspects of the model, for example, to better examine the role of other job resources and demands, as well as the role of other dispositional variables. Furthermore, given the importance of family resources in this context, also in light of the results found, further aspects related to the family should be examined. Moreover, future research should focus more on the family-to-work direction. In fact, while in the work-to-family direction, the role of the variables examined appears to be relatively clear, as for the opposite direction the dynamics and the role of the variables still need to be clarified.

### 5.2. Conclusions and Practical Implications

Despite its limitations, this study contributed to the work-family literature and might provide significant suggestions for potential practical organizational interventions.

Overall, this study contributed to the work-family literature in different ways by:(a)updating and untangling the literature on the topic of WFB, which is still controversial and has, to date, received less empirical attention [8];(b)examining and testing, in Italy, an alternative theoretical model with respect to those proposed in the literature, since some socio-demographic changes, also linked to the Italian context, have implications in terms of work organization, social welfare, and social and family life. In fact, one of the reasons for these changes is due to the fact that in Italy dual-income families are growing rapidly and are becoming the dominant breadwinner kind of families [21];(c)highlighting the differential and indirect effects of conflict and enrichment on WFB through job and family satisfaction;(d)analyzing the moderating role of a dispositional variable within this context as CSE and potential differences due to the moderating effects of this latter construct.

From the results of this study, a fundamental role appears to have also been played by family resources that strongly mitigate conflicts between family and work. It would be feasible for organizations to carry out periodic assessments of the levels of conflict and enrichment in order to guarantee and facilitate greater feelings of satisfaction and balance.

Organizations should effectively and consistently develop and monitor interventions and strategies, of both a formal and informal nature [8].

Hence, organizations should invest time and money in order to promote WFB among workers by providing flexible benefits and resources, for example, by providing support and resources (also coming from colleagues or supervisors) in order to reduce conflict and increase enrichment.

For example, through informal support, improving the knowledge of the positive implications of a work-family culture and highlighting the best practices to promote it. In addition, promoting training activities for supervisors and colleagues to allow them to grow professionally and become more aware of the dynamics relating to work and family contexts.

Indeed, enabling employees to improve and have greater professional development, through training, mentoring, and tutoring, is also of paramount importance in order to increase WFE, as the resources and skills developed in the work domain can also be used in the family domain [21].

Finally, managers, human resource departments, or, more generally, organizations should be more sensitive to the needs of employees and support them by harmonizing and balancing job and family roles, also implementing and controlling the impact of work-family policies. Generally speaking, an environment should be created to help employees achieve this balance, by reducing conflict and improving enrichment. Practically, organizations should encourage and implement new forms of flexibility in work organization, smart working, wellness programs, conciliatory vouchers, and childcare, increasing the use of parental leave, etc.

Overall, in the light of our results from a practical point of view, it would be appropriate to work above all in terms of prevention, including protocols that allow management to intervene in the management of practices and work characteristics related not only to balance, conflict, and enrichment but also to both job and family satisfaction. A key element is certainly a shared support strategy, in which supervisors can help workers to support colleagues, in order to achieve better satisfaction and balance between family and work contexts through, for example, delegation, empowerment, and support resources (see self-efficacy programs).

In addition to the implementation of corporate actions and strategies in the direction of greater WFB, it is also essential to intervene on the set of public tools and policies (welfare, contracts, education, etc.) that are able to increase the so-called cultural and social capital [79]: often the differences and the consequent inequalities in the complex management of family requests and their effect on work, are linked to the set of skills, understood as capital (symbolic, social, and cultural), that the family owns and uses. If the set of lasting dispositions and practical sense in the family (understood as habitus) are able to foster the children’s educational performance [80], then it will also be possible to intervene on policies able to favor and disseminate the elements of social and cultural capital, reporting the best practices in the management of the WFB and the most important factors for WFC.

To conclude, from a practical point of view, improving knowledge of WFB dynamics would allow organizations to both support and help people and their families find a greater balance that would be reflected not only on well-being, individual and family satisfaction, but also on organizational variables, including performance, commitment, and job satisfaction.

This balance-based perspective, also in contrast to conflict-based perspectives, seeks to emphasize advantages in role management and organizational practices. The “work-family balance” construct provides a metaphor, indicating this dimension as a quality emerging from the relationship between the two domains, work and family. This consideration also contrasts the traditional and apparently intuitive idea that work and family relationships can often be in conflict and mutually excluding each other.

## Figures and Tables

**Figure 1 behavsci-10-00140-f001:**
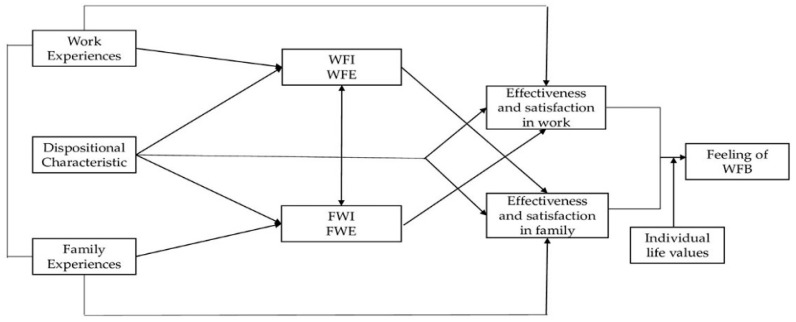
Work-family balance model by Greenhaus and Allen (2011).

**Figure 2 behavsci-10-00140-f002:**
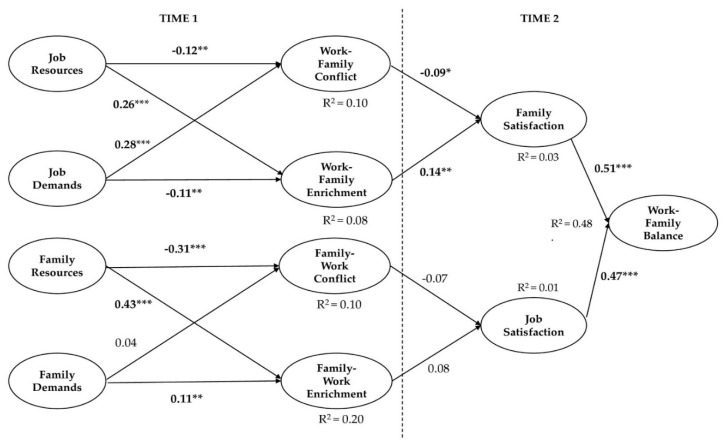
Structural model (standardized coefficients). Note: * *p* < 0.05; ** *p* < 0.01; *** *p* < 0.001.

**Table 1 behavsci-10-00140-t001:** Demographic variables and descriptive statistics.

Demographic Variable	Descriptive Statistics
**Gender**	Men: 263 (46.3%) Women 305 (53.7%)
**Average number of children**	1.45 (SD = 1.06)
**Marital status (married or cohabiting)**	546 (96.1%)
**Educational levels**	Elementary/junior high school certificate 137 (24.1%) High school diploma 238 (41.9%) University degree or higher 193 (34%)
**Employment status**	Permanent employment contract 469 (82.6%) Fixed-term/temporary contract 54 (9.6%) other statuses 44 (6.7%) 1 missing value (0.1%)
**Average organizational tenure (years)**	14.66 (SD = 11.26)
**Average total tenure (years)**	21.74 (SD = 12.17)
**Professional profile**	Blue-collar workers 177 (31.2%)
**Professional sector**	White-collars 314 (55.3%) Managers 54 (9.5%) Professionals or self-employed 21 (3.7) 2 missing values (0.2%) Primary 24 workers (4.2%) Secondary 177 (31.82%) Tertiary 367 (66.4%)

**Table 2 behavsci-10-00140-t002:** Alternative measurement models on study variables.

	χ^2^	df	RMSEA	CFI	SRMR	NNFI
Model 1—one factor	14922.98	1224	0.173	0.34	0.17	0.31
Model 2—complete model	3656.00	1188	0.059	0.93	0.06	0.92

**Table 3 behavsci-10-00140-t003:** Study variables’ descriptive statistics and zero-order correlations.

	M (*SD*)	1	2	3	4	5	6	7	8	9	10	11	12	13	14	15
(1) Workload ^1^	3.33 (0.98)															
(2) Coworkers support	3.70 (0.94)	−0.06														
(3) Supervisor support	4.07 (0.94)	−0.02	0.46 ***													
(4) Job resources	3.90 (0.80)	−0.05	0.85 **	0.85 **												
(5) Family workload ^2^	2.39 (0.10)	0.15 ***	−0.01	−0.02	−0.01											
(6) Emotional family support	4.26 (0.73)	−0.05	0.17 ***	0.15 ***	0.21 **	−0.06										
(7) Instrumental family support	3.88 (0.87)	−0.10 *	0.15 ***	0.11 ***	0.17 **	−0.44 ***	0.47 ***									
(8) General Family support ^3^	3.92 (0.69)	−0.09 *	0.18 ***	0.15 ***	0.20 **	−0.31 ***	0.83 ***	0.88 ***								
(9) Core self-evaluations	3.60 (0.59)	−0.17 ***	0.07	0.19 ***	0.15 **	−0.21 ***	0.19 ***	0.22 ***	0.24 ***							
(10) Work-family Conflict	3.44 (1.6)	0.34 ***	−0.06	−0.12 **	0.11 **	0.11 **	−0.12 **	−0.15 ***	−0.16 ***	−0.28 ***						
(11) Family-work conflict	2.37 (1.3)	0.19 ***	−0.03	−0.09 *	−0.07	0.12 **	−0.28 ***	−0.26 ***	−0.31 ***	−0.30 ***	0.54 ***					
(12) Work-family enrichment	3.86 (0.99)	−0.12 **	0.13 **	0.30 ***	0.25 **	−0.07	0.29 ***	0.11 **	0.23 ***	0.30 ***	−0.17 ***	−0.13 **				
(13) Family-work enrichment	3.88 (0.88)	−0.04	0.09 *	0.20 ***	0.19 **	−0.01	0.40 ***	0.19 ***	0.34 ***	0.29 ***	−0.09 *	−0.14 **	0.58 ***			
(14) Family satisfaction	5.5 (1.2)	−0.04	0.07	0.10 *	0.10 *	−0.16 ***	0.05	0.05	0.06	0.27 ***	−0.10 *	−0.10 *	0.13 **	0.11 **		
(15) Job satisfaction	4.2 (0.89)	−0.03	0.03	0.04	0.04	−0.08 *	0.04	0.10 *	0.08 *	0.20 ***	−0.06	−0.07	0.15 **	0.05	0.35 ***	
(16) Work-family balance	5.70 (0.73)	−0.03	0.03	0.09 *	0.07	−0.16 ***	0.05	0.11 **	0.10 *	0.26 ***	−0.09 *	−0.12 **	0.09 *	0.05	0.59 ***	0.57 ***

Note: * *p* < 0.05, ** *p* < 0.01, *** *p* < 0.001; ^1^ Latent variable: Job demands, ^2^ Latent variable: Family demands, ^3^ Latent variable: Family resources.

**Table 4 behavsci-10-00140-t004:** Direct effects and hypotheses tests.

Effect	β	*p*	Hypothesis Test
Job resources→WFC	−0.12	=0.005	H1a supported
Job resources→WFE	0.26	<0.001	H1b supported
Job demands→WFC	0.28	<0.001	H2a supported
Job demands→WFE	−0.11	=0.005	H2b supported
Family resources→FWC	0.31	<0.001	H3a supported
Family resources→FWE	0.43	<0.001	H3b supported
Family demands→FWC	0.04	Ns	H4a unsupported
Family demands→FWE	0.11	=0.005	H4b supported
WFC→family satisfaction	0.09	=0.05	H5a supported
WFE→family satisfaction	0.14	=0.005	H5b supported
FWC→job satisfaction	−0.07	Ns	H6a unsupported
FWE→job satisfaction	0.08	Ns	H6b unsupported
Family satisfaction→WFB	0.51	<0.001	H7a supported
Job satisfaction→WFB	0.47	<0.001	H7b supported

Note: Ns = Non significant.

**Table 5 behavsci-10-00140-t005:** Moderation effects of CSE.

Effect	
**Job resources→WFC**	Ns
**Job resources→WFE**	Ns
**Job demands→WFC**	Ns
**Job demands→WFE**	Ns
**Family resources→FWC**	S
**Family resources→FWE**	Ns
**Family demands→FWC**	Ns
**Family demands→FWE**	Ns

Note: Ns = Non significant; S = Significant.

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
