# Peer review of "A Time-Lagged Examination of the Greenhaus and Allen Work-Family Balance Model"

_behavsci, 2020, doi:10.3390/bs10090140_

Round 1
Reviewer 1 Report
- This study aims to test work-family balance into an Italian sample, referring to Greenhaus and Allen model. Paper main contributions regard the essential role of work and family resources and work and family demands in managing the dynamics relating to family and work contexts. These constructs can contribute to counterbalancing the perception of incompatibility and conflict, improving both enrichment and feelings of balance generated in both contexts.
- I think that introduction is adequate and that the §2, Theoretical foundation, is widely developed. I suggest, however, to summarize in a table the literature review supporting § 2.1-2.2-2.3-2.4
- I advise the authors to simplify the presentation of the hypotheses, reducing the number, since they are now 11. Maybe this is possible by recalling the initial reference model.
- It is unclear whether the questionnaire at Time 2 is identical to that administered at Time 1. Furthermore, the authors need to clarify why they conducted two surveys three months apart. So in §3.3, they will report the analysis for Time 1 and 2. Finally, they must argue more about what they stated in this paragraph. (§5.1): “Firstly, although the research is time-lagged (twice) and given the nature of the hypotheses of this study, causal inferences cannot be made regarding the relationships between the variables. Given the complexity of the model, to overcome this limitation, future research should preferably focus more on longitudinal or diary studies, or in any case, on a longer lagged design”.
- §3.1 does not describe the administration procedure.
- Authors need to present in a table socio-demographic data (§ 3.1)
- §3.2, authors must specify the following aspects:
- An example of the Coworkers support scale.
- The family workload scale is a shortened version of the original one?
- An example of the Job satisfaction scale.
- §4:
- I suggest indicating in table 2 the eleven latent variables.
- General Family support, does not appear in the text before table 2. Why?
- Rather than reporting Pearson's r, already indicated in Table 2, it's useful to highlight the most significant correlations and then explain the weakly significant and the not significant ones. Similarly for the results described from p.11 line 18 to p.12 line 37.
- I think that integrating §4.2 with §4.1 can facilitate comprehension of the model as a Whole.
- P.13 line 80: you accidentally wrote Allan instead of Allen
- §5:
- “Overall, this study contributed to the work-family literature in different ways: a) by updating and untangling the literature on the topic of WFB, which is still controversial and has, to date, received less empirical attention [8]; b) by examining and testing, in Italy, an alternative theoretical model with respect to those proposed in the literature; c) by highlighting the differential and indirect effects of conflict and enrichment on WFB through job and family satisfaction; d) by analysing the moderating role of a dispositional variable within this context as core self-evaluations and potential differences due to the moderating effects of this latter construct”, this paragraph represents a part of the conclusions e must be inserted in §5.2
- The authors should summarize via a table containing results associated with each hypothesis, points described from line 89 to line 112 of p.13.
- You have to correct references indicating Presti, A.L. in Lo Presti, A.
- Presti, A.L.; Molino, M.; Emanuel, F.; Landolfi, A.; Ghislieri, C. Work-family organizational support as a predictor of work-family conflict, enrichment, and balance: crossover and spillover effects in dual-income couples. Eur. J. Psychol., 2020. 16(1): p. 62-81.
- Presti, A.L.; D’Aloisio, F.; Pluviano, S. With a little help from my family: A mixed-method study on the outcomes of family support and workload. Eur. J. Psychol., 2016. 12(4): p. 584.
- Presti, A.L.; Nonnis, M. Testing the Job Demands-Resources model: Evidence from a sample of Italian employees. TPM: Test., Psychom., Method. in App. Psychol., 2014. 21(1): p. 89-101.
- Emanuel, F.; Molino, M.; Presti, A. L.; Spagnoli, P.; Ghislieri, C. A crossover study from a gender perspective: the relationship between job insecurity, job satisfaction, and partners’ family life satisfaction. Front. Psychol., 2018. 9: p.1481.
- Landolfi, A.; Presti, A.L. A psychometric examination of the work-family balance scale. A multisample study on Italian workers. Curr. Psychol., 2020: p. 1-10.
Reviewer 2 Report
Dear Authors!
This is a valuable contribution addressing a problem of high relevance.
However, there are some minor issues that could/should be improved, or at least clarified, to make the interpretation more evident.
- The measures used in the questionnaire are of core importance. In the Methods section it does not become very clear: were ALL MEASURES taken from standardized questionnaires or some of them were self-compiled by the authors? This should be more clearly formulated in a few sentences. After carefully looking at all references, one can infer that the measures existed before. This does not deduce from the value of the study.
- More importantly: From the presentation of results it does not become clear what the added value of the time-lagged study was (data collection at Time 1 and Time 2). How are these two surveys connected, if no causality is assumed?
- I suggest to articulate more that the family-to-work direction works opposite to expectations, not just in 1 sentence on page 2 after Table 2. In reading the huge amount of evidence and associations, the reader might get lost in them and lose this important statement.
- Some more general inferences – even if carefully formulated due to limited generalizability – regarding the connections between resources, demands, conflict, enrichment, satisfaction and their overall impact on WFB would be welcome at the end of the paper.
Last, three minor issues:
On page 5, the concept of CSE is introduced and explained, however, the words Core self-esteem appear first on page 6 only, this should be put where it is first mentioned as abbreviation.
On the page after Table 2, line 26, FEW is written wrong (FEW). Page 2 after Table 2, lines 48-50, the sentence should be reformulated to make it more clear: …in particular, the association….
In the abstract, the term SEM is not clarified and may be confusing.
I value the work done with the paper. I am sure that just a little bit of clarifying work will make this contribution much more understandable, traceable and it will increase its heuristic value for the reader.
Reviewer 3 Report
I appreciate the opportunity to review this interesting article. In order to improve the contributions of this work and increase the interest of researchers in it, here are a series of suggestions to the authors:
-Follow the APA regulations, for example regarding the minimum number of sentences that a paragraph must contain; the use of acronyms, or amounts at the beginning of a sentence.
-Remove the marks of previous revisions
-Critically assess the limitations of the models used
-reconsider the accent on the Italian sample. As it is in the manuscript, it is expected that what is special about the sample will be justified.
-the work highlights what can be done from the organization with respect to the WLB, interesting to highlight what can be done from the family (or how the family contributes) and its situational variables, its values ​​and its culture with respect to the WLB. Although the authors make an approach to this perspective with family demand and family support, it is recommended to strengthen this perspective by referring to the concept of habitus and cultural family capital (based on Bourdieu's theory), precisely the authors have a work in which these concepts are applied in musical performance (see this work in the journal Music Education Research, 2014). With this, the discussion and conclusions (in which the cultural dimension is mentioned) can also be reinforced.
-the measuring instruments require more and better explanation (authorship of the original scale, language of the original scale, previous applications ...)
-Specify the total number of items to which the participants responded, the order in which the scales were presented, the balance,… as well as other procedural questions that scientific articles should include (see APA recommendations).
-the practical implications of the results obtained should be more and more elaborated. In this regard, the abstract should also make explicit the implications / applications of the knowledge generated.
Reconsider the job title. It is recommended that it be more concise, focusing on what the work contributes.
Author Response
Please see the attachment (for reviewer 3)

Round 2
Reviewer 1 Report
- After the revision of the text, I agree not to include the table showing the studies supporting the hypotheses.
- Overall, I appreciated the changes made to the text and the answers given by the authors.
- I think it is necessary to insert tables 4 and 5, complete with a column reporting, for each row, the corresponding hypothesis, and if each one has been satisfied or not. I confirm that in this way, to simplify the understanding of the text, the following passage from Discussion can be omitted:
“In line with the initial hypotheses, the following significant associations were found: job resources were found to be negatively associated with WFC (H1a) and positively associated with WFE (H1b), as well as job demands were found to be positively associated with WFC (H2a) and negatively associated with WFE (H2b). Furthermore, family resources were negatively associated with FWC (H3a) and positively with FWE (H3b). The present study did not confirm this association between family demands and FWC (H4a).While, as regards the association between family demands and FEW (H4b), contrary to our initial hypothesis, significant positive effects, rather than negative, were found. As for family satisfaction, it was found to be associated negatively with WFC (H5a) and positively with WFE (H5b), while no significant effect had been found between job satisfaction neither with FWC (H6a) nor with FWE (H6b). Furthermore, in line with the reference model [9], both family (H7a) and job satisfaction (H7b) had significant predictive effects on WFB, confirming the initial hypotheses. The indirect mediation effects of family satisfaction between WFE (positively) and WFC (negatively) with WFB were also confirmed. While no effect of mediation of job satisfaction between FWE and FWC with WFB was found. As for dispositional effects, family resources were identified as a negative significant predictor of FWC, with significant moderation effects of CSE , thus providing only partial support for the hypothesis. Therefore, evidence was found for the interactive effect of family resources and CSE on FWC. This is important as it supports the idea that family resources do not have an impact on all individuals in the same way. In fact, compared to subjects with a lower CSE, individuals with a higher CSE, appear to be better equipped with (cognitive and dispositional) resources required to effectively deal with conflicting dynamics, at least in the family-to-work direction. Contrary to the initial hypotheses, no other CSE moderating effects have been found”.
Reviewer 3 Report
The authors have worked on the recommendations of the reviewers. However, some questions must be corrected, basically of form, of the style of scientific writing.
For example, in the Abstract, work-family balance (WFB) should be replaced by Work-Family Balance (WFB) and equation modeling (SEM) by Structural Equation Modeling (SEM). The rules for writing numbers at the beginning of the sentence must also be followed.
In addition, in different parts of the text, it is necessary to make corrections so that the paragraphs include the minimum number of sentences required by regulations, as well as correct other errors such as "confirmatory factor analysis (CFA)".
Rethink the keywords, compare the proposed keywords with the Thesaurus index.
Finally, review / complete references 9, 16 and 49.
In summary, the manuscript is accepted, it should only be revised for style reasons.
